# POST-TRAINING RECOVERY FROM INJECTED BIAS WITH SELF-INFLUENCE

## ABSTRACT

Learning generalized models from biased data with strong spurious correlations to the class label is an important undertaking toward fairness in deep learning. In the absence of any prior knowledge or supervision of bias, recent studies tackle the problem by presuming the bias severity to be sufficiently high and employing a bias-amplified model trained by empirical risk minimization (ERM) to identify and utilize *bias-conflicting* samples that are free of spurious correlations. However, insufficient preciseness in detecting *bias-conflicting* samples results in injecting erroneous signals during training; conversely, it leads to learning malignant biases instead of excluding them. In practice, as the presumption about the magnitude of bias often does not hold, it is important for the model to demonstrate robust performance across a wide spectrum of biases. In this paper, we propose SePT (Self-influence-based Post-Training), a fine-tuning framework leveraging the self-influence score to filter *bias-conflicting* samples, which yields a pivotal subset with significantly diminished spurious correlations. Our method enables the quick recovery of a biased model from learned bias through fine-tuning with minimal friction. In addition, SePT also utilizes the remaining training dataset to adjust the model, thereby maintaining robust performance in situations with weak spurious correlation or even in the absence of it. Experiments on diverse benchmark datasets with a wide range of bias strengths show that SePT is capable of boosting the performance of both bias-injected and state-of-the-art debiased models.

## 1 INTRODUCTION

Deep neural networks have demonstrated remarkable performance in various fields of machine learning tasks such as image recognition (Dosovitskiy et al., 2021), natural language processing (Brown et al., 2020), and speech recognition (Zhang et al., 2020). Under well-curated benchmarks, they are capable of achieving near-human or superhuman performance (He et al., 2015). However, whether these models can still effectively learn with unfiltered real-world data is yet to be fully answered. One of the detrimental artifacts of unfiltered data is *dataset bias* (Torralba & Efros, 2011), where task-irrelevant attributes are spuriously correlated with labels in the curated dataset. Learning from data containing malignant biases inclines the model to rely on exploiting them for the designated task instead of learning the task-related features, resulting in a *biased model* and poor generalization performance (Zhu et al., 2017; Geirhos et al., 2020). For instance, since most seagull images feature the sea as a background, a model may fail to recognize seagulls in different backgrounds, such as meadows and deserts (Sagawa et al., 2020).

To effectively learn from biased datasets, it is important to encourage the model to utilize task-related features rather than malignant bias. A straightforward solution is to utilize explicit supervision or prior knowledge of bias (Kim et al., 2019; Sagawa et al., 2020). Nonetheless, relying on human inspection to alleviate bias can be impractical due to its exorbitant cost and infeasibility in real-world scenarios. Instead, recent studies attempt to glean bias-conflicting samples within the biased trainset by using bias prediction (Liu et al., 2021), loss (Nam et al., 2020; Liu et al., 2023), or gradients (Ahn et al., 2023) obtained from an auxiliary biased model trained with Empirical Risk Minimization (ERM). These samples are then utilized at training to amplify task-related features through loss weighting (Nam et al., 2020) or weighted sampling (Liu et al., 2021), neutralizing the bias. While these approaches can identify bias-conflicting samples to a certain extent, failure to detect them may result in the wrong amplification of bias-aligned samples during training, compromising the task-

relevant attributes. Moreover, they assume that the bias in the trainset is severe enough to induce a strong bias in the ERM-trained auxiliary model. However, such an assumption may not hold in real-world scenarios where the bias is only mildly present, limiting the applicability of the approach.

Reflecting on the problems of training stage intervention, another recent approach involves post-hoc rectification using auxiliary bias information, which is considerably less intrusive compared to the prior approaches. Focusing on the observation that deep neural networks can learn task-related features with ERM even under biased settings (Menon et al., 2021; Kirichenko et al., 2023), they retrain the classification layer to rectify the bias while keeping the feature extractor intact, which is referred to as last layer retraining. However, their effectiveness in the absence of bias supervision or an unbiased validation set is not yet to be fully explored, especially in highly biased datasets.

In this sense, it is required to develop a bias recovery training method that can accurately filter bias-conflicting data. Thus, we propose a post-training method to rectify the injected bias persisting in the model. We first make an attempt to employ the Influence Function (IF), which quantifies the impact of a training sample on the model parameters, in identifying bias-conflicting samples. By measuring Self-Influence (SI) (Koh & Liang, 2017), the influence of a sample on itself through its effect on model parameters, it is possible to detect data that are against the generalization of the model. In this process, directly applying SI does not yield sufficient results. Therefore, we proposed bias-customized self-influence (BCSI) to identify bias-conflicting samples. When the training data is biased, the model would first attempt to generalize on biases, and bias-conflicting samples exhibit larger BCSI scores compared to bias-aligned samples. Based on this observation, we first produce a pivotal subset with significantly diminished spurious correlations by measuring the BCSI of the training samples. Subsequently, we fine-tune a biased model with a few number of iterations, effectively rectifying the bias present in the model even after being debiased by previous approaches. Furthermore, SePT effectively rectifies the existing methods in low-bias scenarios by utilizing both the pivotal set and the remaining samples in the trainset.

Our contributions are threefold:

- We propose bias-conditioned self-influence (BCSI) to filter bias-conflicting samples within the trainset with greater accuracy.

- We propose a novel fine-tuning scheme capable of quickly recovering biased models, even those that have undergone state-of-the-art debiasing techniques.

- Our method not only enhances performance in highly biased settings but also rectifies the existing methods that struggle in low-bias scenarios.

## 2 BACKGROUND

### 2.1 LEARNING FROM BIASED DATA

We consider a supervised learning setting with training data $T := \{z_n\}_{n=1}^N$ sampled from the data distribution $\mathbf{Z} := (X, Y)$, where the input $X$ is comprised of $X = (S, B, O)$ where $S$ is the task-relevant signal, $B$ is a task-irrelevant bias, and $O$ is the other task-independent feature. Also, $Y$ is the target label of the task, where the label is $y \in \{1, \ldots, C\}$. When the dataset is unbiased, ideally, a model learns to predict the target label using the task-relevant signal: $P_\theta(Y|X) = P_\theta(Y|S, B, O) = P_\theta(Y|S)$. However, when the dataset is biased, the task-irrelevant bias $B$ is highly correlated with the task-relevant features $S$ with probability $p_y$, i.e., $P(B = b_y|S = s_y) = p_y$, where $p_y \geq \frac{1}{C}$. Under this relationship, a data sample $x = (s, b, o)$ is *bias-aligned* if $(b = b_y) \wedge (s = s_y)$ and, *bias-conflicting* otherwise.[1] For example, a sample image $x = (s, b, o)$ of the number $0$ in a handwritten digit dataset contains the shape signal $s$, which is directly related to the label $y$, and the other unrelated information such as the color ($b$) of the digit or the background ($o$). However, if all the images containing the digit $0$ in the trainset are colored red, then $b$ is correlated with the digit shape $s$, resulting in bias alignment. When $B$ is easier to learn than $S$ for a model, the model may discover a shortcut solution to the given task, learning to predict $P_\theta(Y|X) = P(Y|B)$ instead of $P_\theta(Y|X) = P(Y|S)$. However, debiasing a model inclines the model towards learning the true task-signal relationship $P_\theta(Y|X) \approx P(Y|S)$.

---

[1]Here, $\wedge$ denotes the logical conjunction.

## 2.2 INFLUENCE FUNCTIONS

The Influence Function (IF; Koh & Liang (2017)) estimates and interprets the effect of each sample in the trainset with respect to the model's prediction. A naive approach for assessing the influence on model predictions is excluding the data point from the trainset and comparing differences in performance, referred to as leave-one-out (LOO) retaining. However, performing LOO retraining for all samples is computationally expensive; instead, an approximated method called influence functions has been introduced as an alternative.

Here, we review the formal definition of influence function. Given a training dataset $T := \{z_n\}_{n=1}^{N}$ where $z_n = (x_n, y_n)$, the model parameters $\theta$ are learned using $T$ with a loss function $\mathcal{L}$:

$$\theta^* := \arg\min_{\theta} \mathcal{L}(T, \theta) = \arg\min_{\theta} \sum_{n=1}^{N} \ell(z_n, \theta) \tag{1}$$

where $\ell(z_n, \theta) := -\log(P_\theta(y_n|x_n))$ is the cross-entropy loss for $z_n$.

To measure the impact of a single training point $z$ on the model parameters, we consider the retrained parameter $\theta^*_{z,\epsilon}$ obtained by up-weighting the loss of $z$ by $\epsilon$:

$$\theta^*_{z,\epsilon} = \arg\min_{\theta}(\mathcal{L}(T, \theta) + \epsilon \cdot \ell(z, \theta)). \tag{2}$$

Then, Influence Function, the impact of $z$ on another sample $z'$, is defined as the deviation of the retrained loss $\ell(z', \theta^*_{z,\epsilon})$ from the original loss $\ell(z', \theta^*)$:

$$\mathcal{I}_\epsilon(z, z') := \ell(z', \theta^*_{z,\epsilon}) - \ell(z', \theta^*) \tag{3}$$

For infinitesimally small $\epsilon$, we have

$$\mathcal{I}(z, z') := \left.\frac{d\mathcal{I}_\epsilon(z, z')}{d\epsilon}\right|_{\epsilon=0} = \nabla_\theta \ell(z', \theta^*)^\top H^{-1} \nabla_\theta \ell(z, \theta^*) \tag{4}$$

where $H := \nabla_\theta^2 \mathcal{L}(T, \theta^*) \in \mathbb{R}^{P \times P}$ is the Hessian of the loss function with respect to the model parameters at $\theta^*$. Intuitively, the influence $\mathcal{I}(z, z')$ measures the effect of $z$ on $z'$ through the learning process of the model parameters. Note that IF is computed once a model has converged since Equation 4 holds when the average of the gradient norm of the trainset is small enough.

Self-influence is introduced as the influence of $z$ calculated on itself:

$$\mathcal{I}_{\texttt{self}}(z) := \nabla_\theta \ell(z, \theta^*)^\top H^{-1} \nabla_\theta \ell(z, \theta^*), \tag{5}$$

which approximates the difference in loss of $z$ when $z$ itself is excluded from training. This metric is effectively used in detecting data with noisy labels (Koh & Liang, 2017) and finding important samples in data pruning (Yang et al., 2023). A high self-influence score indicates that if a sample was omitted from the trainset, making accurate predictions for that sample would become challenging. In other words, the sample contains distinctive information from the majority of the trainset. This characteristic of self-influence enables the detection of samples that cannot be explained straightforwardly using the dominant feature-label relationship learned by the model. For example, Recent studies leverage this characteristic of influence scores to handle the mislabeled samples in the noisy label settings by identifying and removing/relabeling the mislabeled training samples (Koh & Liang, 2017; Ting & Brochu, 2018; Wang et al., 2018; 2020; Kong et al., 2022). Moreover, the influence score can be utilized to select important samples in data pruning for efficient training (Sorscher et al., 2022; Yang et al., 2023). These findings have inspired us to propose using influence scores to identify bias-conflicting samples in a biased dataset, as outlined in Section 3.1.

## 3 SELF-INFLUENCE BASED POST-TRAINING (SEPT)

In this section, we propose Self-Influence based Post-Training (SePT), a debiasing framework that first detects bias-conflicting samples via self-influence and remedies a biased model via post-hoc fine-tuning. In Section 3.1, we show that the direct application of Self-Influence (SI) is not effective in detecting bias-conflicting samples. Based on this comprehensive study, we introduce a modified version of SI, called Bias-Customized Self-Influence (BCSI), which demonstrates effectiveness in

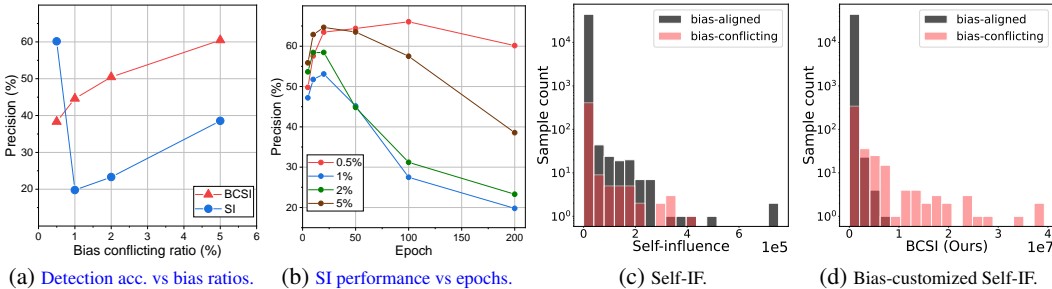

(a) Detection acc. vs bias ratios. (b) SI performance vs epochs. (c) Self-IF. (d) Bias-customized Self-IF.

Figure 1: A comprehensive analysis of Self-Influence (SI) and our Bias-Customized Self-Influence (BCSI) in detecting bias-conflicting (Section 2.1) samples across varying bias ratios. Figure 1(a) shows the detection precision of SI and BCSI across various ratios of bias-conflicting samples for CIFAR10C. Figure 1(b) depicts the detection precision of SI across training epochs for different ratios of bias-conflicting samples. In Figure 1(c) and 1(d), each bar indicates the number of samples within a specific range in CIFAR10C (1%).

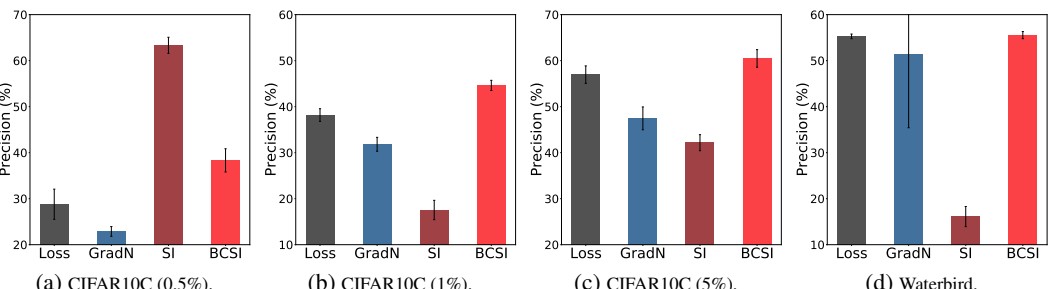

(a) CIFAR10C (0.5%). (b) CIFAR10C (1%). (c) CIFAR10C (5%). (d) Waterbird.

Figure 2: A performance comparison of Loss, Gradient Norm, Self-Influence (SI) and our Bias-Customized Self-Influence (BCSI). The average precision of loss value, gradient norm, SI, and BCSI are presented in bars, with the error bars indicating the standard error across three repetitions.

detection. Subsequently, using BCSI, we successfully identify and construct a concentrated pivotal subset characterized by a high proportion of bias-conflicting samples. In Section 3.2, to effectively use the pivotal subset to remedy biased models, we investigate the efficacy of the last-layer retraining and find that this technique is not as effective unless the dataset exhibits a significantly high ratio of bias-conflicting samples. To this end, we propose a fast and lightweight post-hoc fine-tuning scheme to recover the biased model using the pivotal subset. The overall pipeline of SePT is described in Figure 3.

## 3.1 FILTERING BIAS-CONFLICTING SAMPLES WITH SELF-INFLUENCE

**Understanding the limitations of self-influence in identifying bias-conflicting samples.** In Section 2.2, we discussed the capability of SI to identify samples that contrast to dominant features learned by the pre-trained model such as mislabeled samples (Wang et al., 2018; Kong et al., 2022). Since bias-conflicting samples also conflict with the dominant malignant bias features, SI can be considered as a metric to detect bias-conflicting samples in biased datasets. However, we observe that the direct application of SI to biased datasets is not effective. Figure 1(a) demonstrates the detection performance of SI against various ratios of bias-conflicting samples, using the ground truth count of bias-conflicting samples. Notably, the detection precision of SI is low, falling below 40%, except in the extreme case of 0.5%.

The reason why SI underperforms in biased datasets is that bias-conflicting samples possess correct task-related features, unlike mislabeled samples. In the case of mislabeled samples, which are erroneously labeled as their name implies, they strongly counteract the dominant features of the pre-trained model thereby separable by self-influence. On the other hand, bias-conflicting samples, containing task-related features but under-prioritized in training, differ from the dominant features but do not counteract them. In other words, in a noisy labeled setting, the mislabeled sample's feature is incompatible with the dominant feature, whereas, in a biased setting, the bias-conflicting sample's feature is not only compatible, but ideally, both should be utilized. This characteristic of bias-conflicting samples makes it harder for SI to separate bias-conflicting samples.

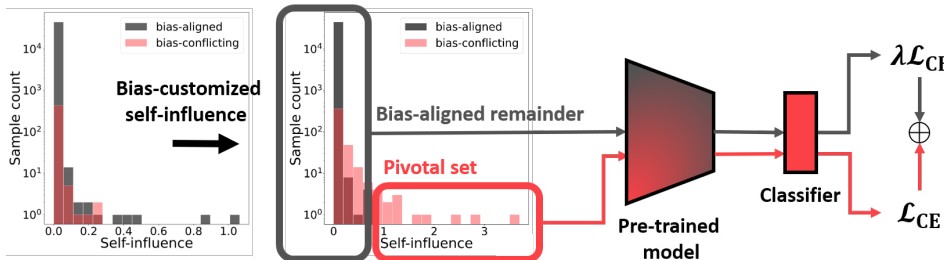

Figure 3: Overview of our framework (SePT). SePT computes the self-influence of the training data using a biased model and then constructs a pivotal set where bias-conflicting samples form a majority. SePT then initializes the last layer of a pre-trained model and trains it using the pivotal set and the remaining data.

For these reasons, as shown in Figure 1(a), in highly malignant scenarios with a bias-conflicting ratio of only 0.5%, self-influence can effectively discriminate, but as the ratio increases, detection performance declines since the model has learned more of the task-related features of bias-conflicting samples for classification. Moreover, Figure 1(b) demonstrates a significant decline as the training epochs increase, due to the model learning the task-related features of bias-conflicting samples.

**Adapting SI to suit biased datasets.** Motivated by our observations, we propose Bias-Customized Self-Influence (BCSI) to restrict the pre-trained model to learn task-related features of bias-conflicting samples.. Based on the observation of Nam et al. (2020) where the loss of bias-aligned data decreases in the early stage of training, we use Generalized Cross Entropy (GCE) (Zhang & Sabuncu, 2018) to induce models to exploit easier-to-learn bias-aligned data, thereby improving detection precision. Furthermore, based on the findings of Frankle et al. (2020) that the primary directions of the model's parameter weights had already been learned during the iteration 500 to 2,000, we train ResNet18 (He et al., 2016) for only five epochs to achieve better sample separation. We exploit the model trained under the aforementioned conditions to employ SI as a means of filtering bias-conflicting samples.

We now validate the capability of BCSI to detect bias-conflicting samples. In Figure 1(a), BCSI shows performance advantages in detection precision compared to conventional SI. In Figure 1(d), there is a noticeable tendency in which bias-conflicting samples exhibit larger scores compared to bias-aligned samples. These findings suggest that BCSI is capable of serving as an effective indicator for detecting bias-conflicting samples within a biased trainset. This trend is also observed in other biased datasets, as shown in Appendix A. To further validate the effectiveness of bias-customized self-influence (BCSI), we compare its average precision with those of loss and gradient norms. In Figure 2(a)-2(d), we present detection precision of loss value, gradient norm, self-influence, and bias-customized self-influence (the detailed settings are described in Appendix B). The results for other datasets are provided in Appendix B. BCSI exhibits dominant performance or comparable precision with other metrics, which is consistent with the findings in previous applications of self-influence (Koh & Liang, 2017; Yang et al., 2023). In noisy label handling, self-influence outperforms the loss values (Koh & Liang, 2017). In data pruning, the selection rules based on naive self-influence are more robust at high pruning ratios compared to the loss-based and gradient-based approaches (Yang et al., 2023).

**Influence-based filtering method.** We now introduce a filtering method to identify bias-conflicting data. To calculate the SI of the trainset, we randomly initialize a model with an identical architecture. By training the model with biased data using GCE for five epochs, we obtain an amplified biased model. Using this model, we compute the SI of the trainset with Equation 5 and rank them in descending order. Since calculating $H^{-1} := (\nabla_\theta^2 L(S, \theta^*))^{-1}$ is generally intractable for deep neural networks due to their extensive number of parameters, we approximately calculate $H^{-1}$ and the loss gradient of the sample $z$, $\nabla_\theta \ell(z, \theta^*)$, of the last layer of the network following the convention (Koh & Liang, 2017; Pruthi et al., 2020). With the obtained SI, we select the top-$k$ subset of samples from each class to form a pivotal subset of bias-conflicting samples as follows: $\mathbf{Z}_\mathrm{P} = \bigcup_{c=1}^{C} \{z_{\mathrm{BCSI\text{-}rank}(m,c)}\}_{m=1}^{k}$, where $C$ is the number of classes and BCSI-rank$(n,c)$ is the dataset index of the $n$-th training sample of class $c$ sorted by bias-customized self-influence.

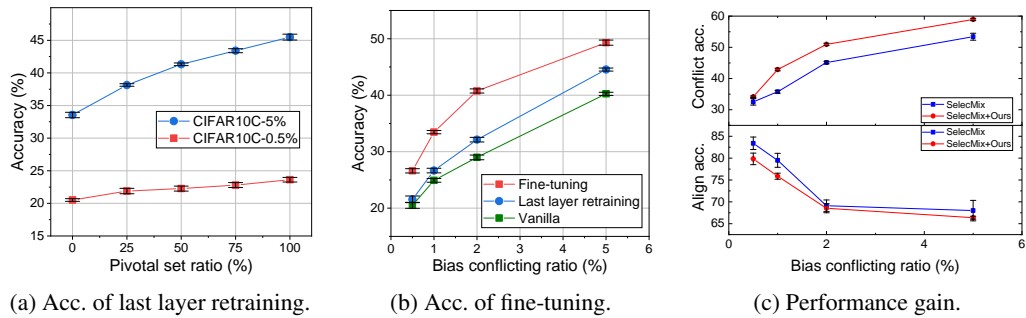

(a) Acc. of last layer retraining.  (b) Acc. of fine-tuning.  (c) Performance gain.

Figure 4: The figures depict performance under varying bias-conflicting ratios. Figure 4(a) shows the accuracy for last layer retraining across varying bias ratios in pivotal sets. Figure 4(b) depicts performance changes of last layer retraining and fine-tuning under diverse bias ratios. In Figure 4(c), our performance gains are provided.

In the experiments, for robustness to random initialization, we repeat this process three times and use the pivotal set as the intersection of the sets obtained through repetition. Since we only train models for a few epochs, this iterative approach requires negligible cost compared to full training of the additional model Nam et al. (2020); Lee et al. (2021); Hwang et al. (2022). The detailed filtering process is provided in Algorithm 1. As a result, we confirmed that the filtering process is capable of constructing a pivotal subset with a high ratio of bias-conflicting samples from a highly biased trainset. Specifically, we observed an increase in the ratio for each dataset in Appendix D.

## 3.2 POST-TRAINING RECOVERY FROM BIAS VIA FINE-TUNING

**Fine-tuning with the pivotal subset.** After constructing the pivotal subset using SI (Section 3.1), we explore options to utilize the acquired set for debiasing. A conventional approach is to directly intervene during model training (Sagawa et al., 2020; Nam et al., 2020). However, training a model directly leveraging a small subset most likely results in overfitting. An inert alternative to this goal is last-layer retraining. Kirichenko et al. (2023) demonstrated that retraining the last layer using an unbiased validation while keeping the well-learned feature extractor frozen can rectify a biased model. However, the pivotal set is not a validation set nor it is perfectly unbiased. Thus, we investigate the feasibility of last-layer training in the absence of an unbiased validation set, which represents a more practical scenario. Considering the observation in Section 3.1, the acquisition of an ideal set consisting of only bias-conflicting samples is challenging in highly biased settings. Therefore, we evaluate the efficacy of last-layer retraining with respect to the degree of the bias-conflicting ratio of a training subset used for retraining. Figure 4(a) shows that a pivotal set with a significantly high bias-conflicting ratio is required to resolve a biased model with last layer retraining alone. However, re-initializing the classifier layer and fine-tuning the entire model yields better performance in Figure 4(b). This trend is more distinctive in *high bias* regimes, where the premise of a well-learned feature extractor may not hold. Therefore, we opt for a fine-tuning method to recover a biased model through post-training recovery. Note that fine-tuning even including the construction of a pivotal set demands less than half the time of full training, as we only train pre-trained models for a few iterations. The detailed comparison of computational costs is suggested in Appendix E.

**Handling weak spurious correlations with counterweighting.** On the opposite end, a significant aspect of dataset bias is when the spurious correlation is *weak*, or even *unbiased*. In real-world scenarios, the robustness of a debiasing method becomes crucial, even in the absence of prior knowledge about the severity of bias. A major pitfall of previous methods is their assumption that the trainset contains a sufficient amount of biased samples, which in turn produces a biased ERM-trained model to be used against itself for debiasing. However, when this assumption is unreliable or invalid as in learning from an unbiased dataset, the scheme may backfire and amplify the malignant bias primarily due to the unbiased ERM-trained model.

To address these concerns, we leverage not only the pivotal subset but also the remaining samples during the fine-tuning stage in order to incorporate the task-related features contained in both bias-conflicting and bias-aligned samples. Specifically, we formulate a counterweight cross-entropy loss by drawing a mini-batch from the remaining trainset. Finally, we train the model using both the

Table 1: The average and the standard error of accuracy over three runs. *Ours* indicates SePT applied to a model initially trained with the prefix method. The best accuracy is annotated in **bold**. ✓indicates that a given method uses bias information while ✗denotes that a given model does not use any bias information.

| Method | Bias Info | CMNIST | | | | CIFAR10C | | | | BFFHQ |
|---|---|---|---|---|---|---|---|---|---|---|
| | | 0.5% | 1% | 2% | 5% | 0.5% | 1% | 2% | 5% | 0.5% |
| GroupDRO | ✓ | 63.12 | 68.78 | 76.30 | 84.20 | 33.44 | 38.30 | 45.81 | 57.32 | 54.80 |
| Vanilla | ✗ | $38.92_{\pm0.74}$ | $56.81_{\pm0.45}$ | $69.19_{\pm0.85}$ | $86.00_{\pm0.35}$ | $20.50_{\pm0.54}$ | $24.91_{\pm0.33}$ | $28.99_{\pm0.42}$ | $40.24_{\pm0.28}$ | $53.53_{\pm2.05}$ |
| ReBias | ✓ | $70.47_{\pm1.84}$ | $87.40_{\pm0.78}$ | $92.91_{\pm0.15}$ | $\mathbf{96.96}_{\pm0.04}$ | $22.27_{\pm0.41}$ | $25.72_{\pm0.20}$ | $31.66_{\pm0.43}$ | $43.43_{\pm0.41}$ | $56.80_{\pm1.56}$ |
| LfF | ✗ | $66.53_{\pm1.24}$ | $78.10_{\pm1.97}$ | $74.69_{\pm4.20}$ | $76.72_{\pm0.94}$ | $25.28_{\pm2.89}$ | $31.15_{\pm1.67}$ | $38.64_{\pm0.39}$ | $46.15_{\pm0.54}$ | $55.33_{\pm2.69}$ |
| DFA | ✗ | $\mathbf{89.64}_{\pm1.40}$ | $\mathbf{94.60}_{\pm0.81}$ | $91.69_{\pm1.53}$ | $95.59_{\pm0.43}$ | $27.13_{\pm1.66}$ | $31.26_{\pm2.71}$ | $37.96_{\pm0.71}$ | $44.99_{\pm0.84}$ | $52.07_{\pm1.91}$ |
| BiaSwap | ✗ | 85.76 | 83.74 | 85.29 | 90.85 | 29.11 | 32.54 | 35.25 | 41.62 | - |
| BPA | ✗ | $54.52_{\pm3.39}$ | $72.63_{\pm0.27}$ | $78.52_{\pm0.59}$ | $85.30_{\pm0.93}$ | $25.50_{\pm1.03}$ | $26.86_{\pm0.69}$ | $27.47_{\pm1.46}$ | $34.29_{\pm2.20}$ | $51.40_{\pm2.98}$ |
| SelecMix | ✗ | $52.60_{\pm0.65}$ | $72.16_{\pm0.79}$ | $80.77_{\pm0.77}$ | $86.86_{\pm3.03}$ | $37.63_{\pm0.81}$ | $40.14_{\pm0.42}$ | $47.54_{\pm0.59}$ | $54.86_{\pm0.76}$ | $63.07_{\pm2.32}$ |
| **Ours**+Vanilla | ✗ | $42.09_{\pm0.72}$ | $62.38_{\pm0.79}$ | $74.34_{\pm0.82}$ | $86.00_{\pm0.35}$ | $26.61_{\pm0.38}$ | $33.47_{\pm0.29}$ | $40.75_{\pm0.37}$ | $49.30_{\pm0.46}$ | $56.00_{\pm1.07}$ |
| **Ours**+LfF | ✗ | $58.84_{\pm2.36}$ | $72.69_{\pm2.25}$ | $79.59_{\pm0.36}$ | $84.78_{\pm0.20}$ | $27.63_{\pm1.00}$ | $35.29_{\pm1.21}$ | $43.36_{\pm0.78}$ | $51.95_{\pm0.29}$ | $57.13_{\pm2.46}$ |
| **Ours**+DFA | ✗ | $76.80_{\pm3.09}$ | $91.17_{\pm1.22}$ | $\mathbf{93.08}_{\pm0.59}$ | $96.21_{\pm0.53}$ | $25.66_{\pm0.85}$ | $33.53_{\pm2.01}$ | $42.80_{\pm0.81}$ | $52.61_{\pm0.54}$ | $56.60_{\pm2.83}$ |
| **Ours**+SelecMix | ✗ | $51.98_{\pm0.49}$ | $71.62_{\pm0.96}$ | $80.79_{\pm0.60}$ | $87.48_{\pm2.52}$ | $\mathbf{38.74}_{\pm0.36}$ | $\mathbf{46.18}_{\pm0.33}$ | $\mathbf{52.70}_{\pm0.40}$ | $\mathbf{59.66}_{\pm0.31}$ | $\mathbf{65.80}_{\pm3.12}$ |

Table 2: Performance of baselines and SePT on low-bias regimes.

| Method | CIFAR10C | | | | |
|---|---|---|---|---|---|
| | 20% | 30% | 50% | 70% | 90%(unbiased) |
| Vanilla | $59.47_{\pm0.59}$ | $65.64_{\pm0.51}$ | $71.33_{\pm0.09}$ | $\mathbf{74.90}_{\pm0.25}$ | $\mathbf{76.03}_{\pm0.26}$ |
| LfF | $59.78_{\pm0.85}$ | $60.56_{\pm0.96}$ | $60.35_{\pm0.37}$ | $62.52_{\pm0.49}$ | $63.42_{\pm0.63}$ |
| DFA | $60.34_{\pm0.46}$ | $64.24_{\pm0.44}$ | $65.97_{\pm1.80}$ | $64.97_{\pm0.20}$ | $66.59_{\pm5.20}$ |
| SelecMix | $62.05_{\pm1.26}$ | $62.17_{\pm0.35}$ | $62.52_{\pm1.54}$ | $66.23_{\pm0.09}$ | $65.81_{\pm0.96}$ |
| **Ours**+Vanilla | $62.78_{\pm0.67}$ | $65.61_{\pm0.77}$ | $70.61_{\pm0.62}$ | $73.20_{\pm0.35}$ | $73.57_{\pm0.16}$ |
| **Ours**+LfF | $64.46_{\pm0.29}$ | $64.40_{\pm0.27}$ | $65.82_{\pm0.15}$ | $67.29_{\pm0.17}$ | $68.15_{\pm0.76}$ |
| **Ours**+DFA | $66.30_{\pm0.48}$ | $\mathbf{68.13}_{\pm0.45}$ | $\mathbf{72.79}_{\pm0.38}$ | $73.56_{\pm0.15}$ | $70.36_{\pm4.08}$ |
| **Ours**+SelecMix | $\mathbf{66.67}_{\pm0.43}$ | $64.51_{\pm1.44}$ | $66.45_{\pm0.28}$ | $69.97_{\pm0.21}$ | $69.29_{\pm0.75}$ |

Table 3: Acc. on Waterbird, NICO

| Method | Waterbird | NICO |
|---|---|---|
| Vanilla | $68.74_{\pm2.65}$ | $39.56_{\pm1.77}$ |
| LfF | $75.27_{\pm2.12}$ | $34.56_{\pm1.47}$ |
| DFA | $77.57_{\pm1.60}$ | $44.59_{\pm0.33}$ |
| SelecMix | OOM | $33.87_{\pm1.27}$ |
| **Ours**+Vanilla | $87.64_{\pm1.30}$ | $43.54_{\pm0.50}$ |
| **Ours**+LfF | $\mathbf{87.85}_{\pm0.68}$ | $40.18_{\pm0.91}$ |
| **Ours**+DFA | $87.12_{\pm0.68}$ | $\mathbf{45.69}_{\pm1.12}$ |
| **Ours**+SelecMix | OOM | $44.33_{\pm0.55}$ |

cross-entropy loss on the pivotal subset and the counterweight loss on the remaining trainset:

$$\mathcal{L}(\mathbf{Z}_\text{P}, \mathbf{Z}_\text{R}) := \mathcal{L}_\text{CE}(\mathbf{Z}_\text{P}) + \lambda\mathcal{L}_\text{CE}(\mathbf{Z}_\text{S}) \tag{6}$$

where $\mathbf{Z}_\text{P}$ is the pivotal subset, $\mathbf{Z}_\text{S} \sim \mathbf{Z} \setminus \mathbf{Z}_\text{P}$ a randomly drawn mini-batch from the remaining trainset, and $\mathcal{L}_\text{CE}$ is the mean cross-entropy loss. Note that we put $\lambda = 0.1$ for all experiments. The overall process is described in Algorithm 2.

# 4 EXPERIMENTS

In this section, we present a series of experiments in which we apply our method to models trained with ERM and existing debiasing approaches, including the current state-of-the-art, to demonstrate the effectiveness of SePT. We validate our method and its individual components following prior conventions. Below, we provide a brief overview of our experimental setting in Section 4.1, followed by empirical results and detailed analyses presented in Section 4.2, Section 4.3, and 4.4.

## 4.1 EXPERIMENTAL SETTINGS

**Datasets.** For fair evaluation, we follow the conventions of using benchmark biased datasets (Nam et al., 2020). Colored MNIST dataset (CMNIST) is a synthetically modified MNIST (Deng, 2012), where the labels are correlated with colors. We conduct benchmarks on bias ratios of $r \in \{0.5, 0.1, 0.2, 5\}$. CIFAR10C is a synthetically modified CIFAR10 (Krizhevsky et al., 2009) dataset with common corruptions. To test our method in low-bias scenarios, we expand our scope and conduct experiments with bias ratios $r \in \{0.5, 0.1, 0.2, 5, 20, 30, 50, 70, 90(\text{unbiased})\}$. Biased FFHQ (BFFHQ) (Lee et al., 2021) dataset is a curated Flickr-Faces-HQ (FFHQ) (Karras et al., 2019)

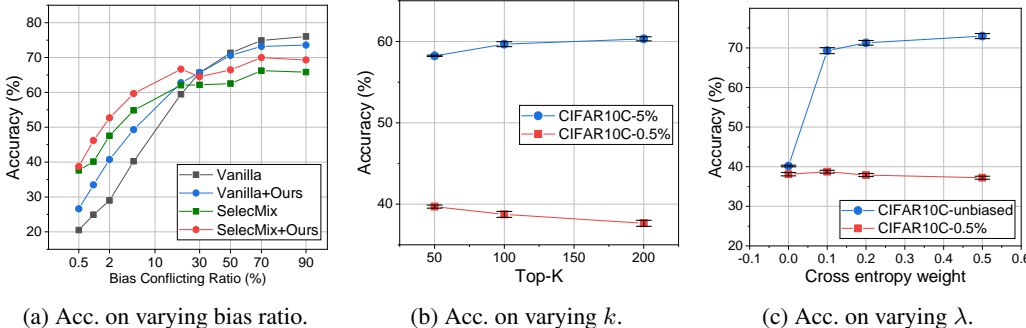

(a) Acc. on varying bias ratio.   (b) Acc. on varying $k$.   (c) Acc. on varying $\lambda$.

Figure 5: Performance of SePT under varying conditions on CIFAR10C. Figure 5(a) displays the test accuracy for SelecMix and SePT at different bias ratios. Figure 5(b) and Figure 5(c) depict the unbiased evaluation under varying the size of the pivotal set and $\lambda$, respectively.

dataset, which consists of facial images where ages and genders exhibit spurious correlation. The Waterbirds dataset (Wah et al., 2011) consists of bird images, to classify bird types, but their backgrounds are correlated with bird types. Non-I.I.D. Image dataset with Contexts (NICO) (He et al., 2021) is a natural image dataset for out-of-distribution classification. We follow the setting of Wang et al. (2021), inducing long-tailed bias proportions within each class, simulating diverse bias ratios in a single benchmark dataset. A detailed description of these datasets is provided in Appendix H.

**Baselines.** We validate SePT by combining various debiasing approaches. GroupDRO (Sagawa et al., 2020) uses group labels to debias. ReBias (Bahng et al., 2020) uses an auxiliary model expertise on specific bias. LfF (Nam et al., 2020) detects bias-conflicting samples based on the assumption that trainset is highly baised. DFA (Lee et al., 2021) and BiaSwap (Kim et al., 2021) augment bias-conflicting samples. BPA (Seo et al., 2022) utilizes a clustering method to identify pseudo-attributes. SelecMix (Hwang et al., 2022) identifies and mixes a bias-contradicting pair within the same class while detecting and mixing a bias-aligned pair from different classes. Note that we adopt SelecMix+LfF rather than SelecMix since SelecMix+LfF exhibits superior performance than SelecMix (Hwang et al., 2022). A detailed explanation for baselines is suggested in Appendix H.2.

**Evaluation protocol.** Following other baselines, we calculate the accuracy for unbiased test sets in CMNIST and CIFAR10C. However, we evaluate minority-group accuracy in BFFHQ, and the worst-group accuracy in Waterbird. Note that we use the models from the final epoch for all experiments to compute performance. A detailed experimental setting is suggested in Appendix H.

## 4.2 RESULTS IN HIGHLY BIASED SCENARIOS

We evaluate SePT to measure the degree of recovery of baseline models when combined with ours on benchmark datasets. In Table 1, we significantly enhance the performance of baselines on the majority of datasets under various experimental settings. To the best of our knowledge, Ours+SelecMix achieves state-of-the-art accuracy on CIFAR10C. Interestingly, we observe that performance gain is larger as the ratio of bias-conflicting samples increases in CIFAR10C. We conjecture that fine-tuning becomes more effective in CIFAR10C (2%) and (5%) since the bias-conflicting sample purity of the pivotal set increases, as shown in Section 3.1. For CMNIST, there are decreases after combining our methods. In Table 5, low detection precision induces performance drop.

## 4.3 RESULTS IN LOW-BIAS SCENARIOS

Since the baseline methods intensify learning signals of bias-conflicting samples strongly, these methods would likely fail in mildly biased datasets. We validate the baselines on CIFAR10C under various ratios of bias-conflicting samples in Table 2 and 3. All the baselines exhibit drastic performance deterioration compared to Vanilla when the bias-conflicting ratio is high. In contrast, our method can significantly rectify remaining biases within a model, even in mildly biased datasets except for Vanilla. Albeit there is a slight decrease in performance for Vanilla, the accuracy gap is much lower than other baselines. Since the innate nature of fine-tuning can minimize friction by training from pre-trained parameters, our approach can remedy biases within a model in a wider range of bias

ratios, as in Figure 5(a). In Waterbird, training SelecMix is intractable since this method simultaneously trains three models of ResNet50 (He et al., 2016). Note that 'OOM' denotes 'out-of-memory'. The graphs for other methods are provided in Appendix C.

## 4.4 ABLATION STUDY

We examine the sensitivity of hyperparameters such as the number of selected samples per class ($k$) in the pivotal set and the weight for the remaining data in post-training ($\lambda$). In Figure 5(b), there is a slight performance decrease as $k$ increases in CIFAR10C (0.5%). In contrast, the accuracy in CIFAR10C (5%) increases. Since there are a few bias-conflicting samples per class in CIFAR10C (0.5%), additional usage of samples dilutes the ratio of bias-conflicting data in the pivotal set, leading to a performance drop. In Figure 5(c), we observe a marginal accuracy drop as $\lambda$ increases in CIFAR10C (0.5%), CIFAR10C (90%) experiences a performance increase. These results indicate that learning the remaining samples is beneficial in CIFAR10C (90%), fostering the model to capture task-relevant signals. We note that the analysis for intersections is provided in Appendix F.

## 5 RELATED WORK

**Debiasing deep neural networks.** The focus of research on mitigating bias has been modulating task-related information and malignant biases during training. Early works relied on human knowledge through direct supervision or implicit information of bias (Sagawa et al., 2020; Li & Vasconcelos, 2019; Hong & Yang, 2021; Han & Tsvetkov, 2021), which is often impractical due to its cost. To address more practical issues, several studies have focused on identifying and utilizing bias-conflicting samples without relying on human knowledge. These methods can be categorized into three main streams: loss modification, sampling methods, and data augmentation. Loss modification methods (Nam et al., 2020; Liu et al., 2023) amplify the learning signals of (estimated) bias-conflicting samples by modifying the learning objective. Sampling methods (Liu et al., 2021; Ahn et al., 2023) overcome dataset bias by sampling (estimated) bias-conflicting data more frequently. Data augmentation approaches (Lee et al., 2021; Lim et al., 2023; Jung et al., 2023) synthesize samples with various biases distinct from the inherent biases of the original data. Recently, based on the observation that biases in classification layers are severe compared to feature extractors, several approaches focus on rectifying the last layers (Kim et al., 2022; Menon et al., 2021; Kirichenko et al., 2023). Especially, Kirichenko et al. (2023) shows that the model learns both task-related features and spurious correlations and proposes retraining the classification layer using an unbiased validation set. However, leveraging an unbiased validation set is also impractical, as previously mentioned.

**Influence functions.** Influence Function (IF; Koh & Liang (2017)) and its approximations Pruthi et al. (2020); Schioppa et al. (2022) have been utilized in various deep learning tasks by measuring the importance of training samples and the relationship between them. One common application of IF is quantifying memorization by self-influence, which is the increase in loss when a training sample is excluded (Pruthi et al., 2020; Feldman & Zhang, 2020). Similarly, self-influence can be used to identify mislabeled samples in the training dataset since they cannot be reliably recovered once removed. Alternatively, Sorscher et al. (2022) eliminate low self-influence samples for computational efficiency as they can be generalized from other samples when removed. On the other hand, the sign of influence has been utilized to identify whether a training sample is beneficial or harmful by measuring its influence with a validation dataset.

## 6 CONCLUSION

In this paper, we thoroughly examined the tendency of self-influence in a biased dataset. We discovered that simply applying self-influence would not be sufficient to detect bias-conflicting samples. Based on this observation, we introduced a strategy to exploit self-influence in identifying bias-conflicting samples. We also demonstrated that fine-tuning is more effective in a highly biased dataset and suggested an approach to rectify biases within a pre-trained model under any given ratio of bias-conflicting samples. We show that our method consistently enhances the existing debiasing approaches across benchmark datasets under various ratios of bias-conflicting samples.

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

# A DISTRIBUTION OF SELF-INFLUENCE AND BIAS-CUSTOMIZED INFLUENCE

In Figure 1(c) and 1(d) of the main paper, we have shown the influence histogram of naive self-influence and bias-customized self-influence (Ours) for the training set of CIFAR10C (1%). In this section, we show the histograms of self-influence and bias-customized self-influence for the training sets of an extended variety of bias ratios and datasets. Figure 7 shows the influence histograms of CIFAR10C for multiple bias-conflicting ratios. Figure 6 shows the influence histograms for BFFHQ and Waterbirds. In accordance with the main paper, we observe that bias-customized self-influence generally exhibits better separation compared to naive self-influence, deeming it a better option to detect bias-conflicting samples.

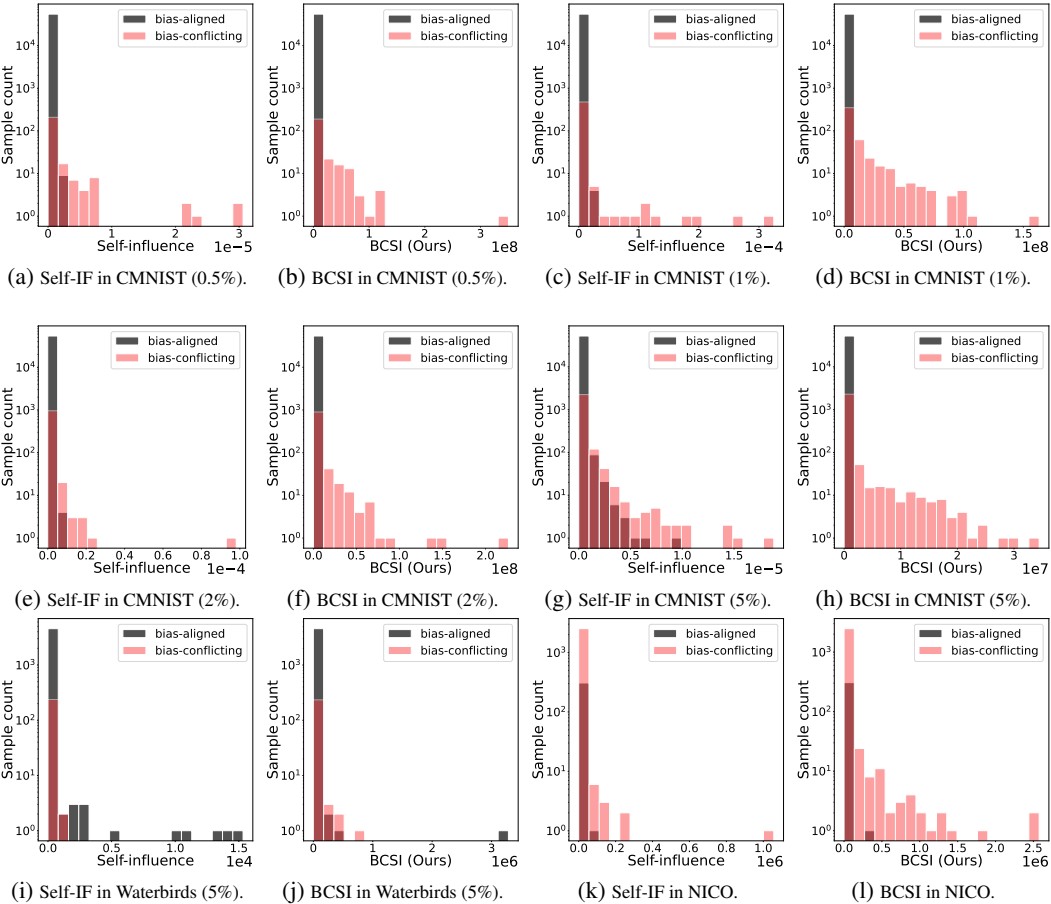

Figure 6: Histogram of self-influence and bias-customized self-influence for CMNIST, Waterbird, and NICO.

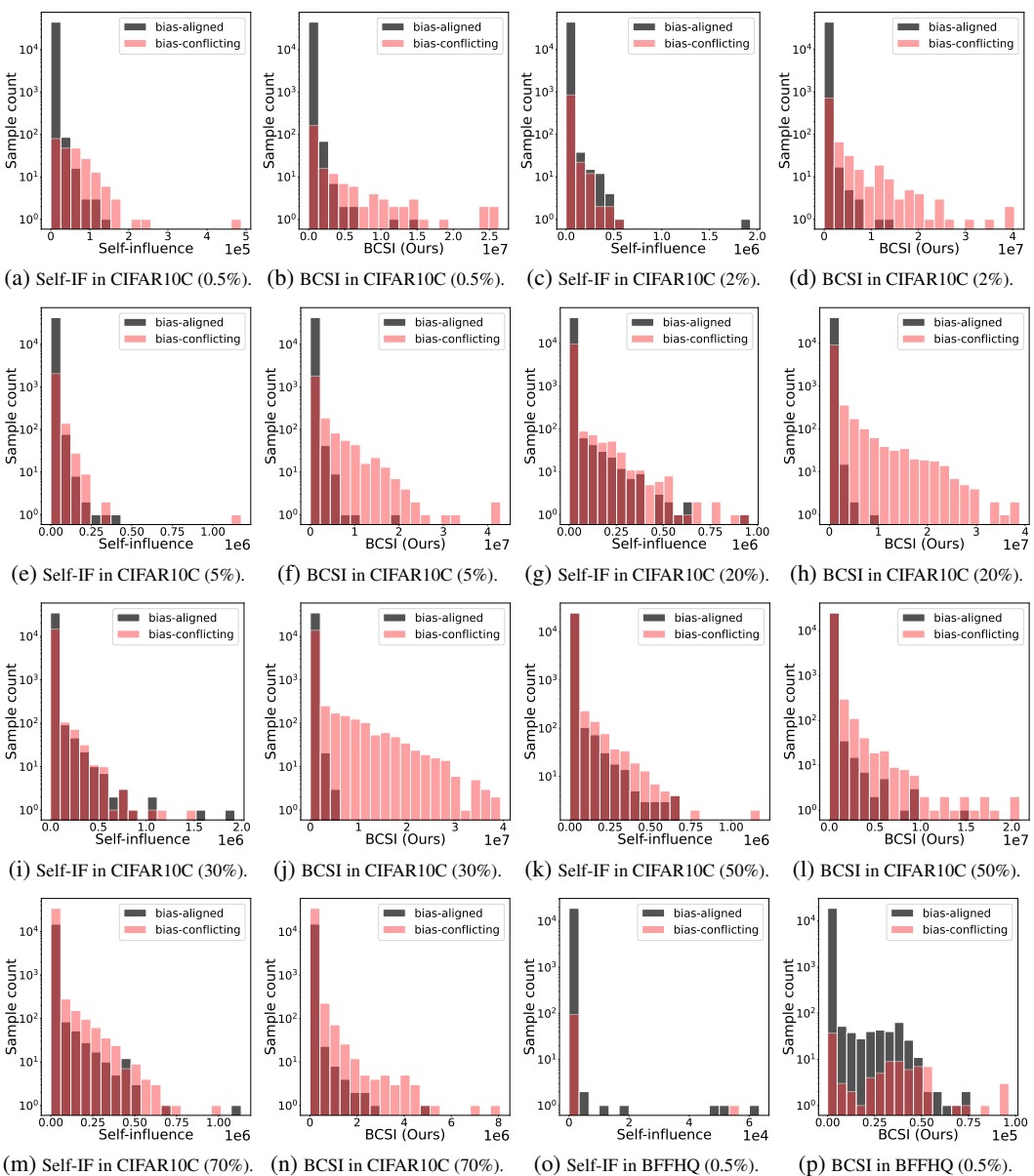

Figure 7: Histogram of self-influence and bias-customized self-influence for the CIFAR10C dataset with varying bias-conflicting ratio and BFFHQ.

# B  DETECTION PRECISION FOR OTHER DATASETS

We now describe the detailed experimental setting used in Figure 2(a)-2(d) of the main paper. We first train ResNet18 (He et al., 2016) for five epochs and then compute loss values, gradient norm, self-influence, and bias-customized self-influence. Note that we only use the last layer when computing gradient norm, self-influence, and bias-customized self-influence. Subsequently, we sort the training data in descending order based on the values obtained by each method, selecting samples ranging from the highest to the $k$-th sample, where $k$ is the number of total bias-conflicting samples in the training set. We then calculate the precision in detecting bias-conflicting samples within the selected data.

To further demonstrate the effectiveness of bias-customized self-influence in detecting bias-conflicting samples, we compare bias-customized self-influence with loss values, gradient norm, and self-influence on other datasets including CMNIST (0.5%, 1%, 2%, 5%), CIFAR10C (2%, 20%, 30%, 50%, 70%, 90%), BFFHQ, and NICO. As shown in Figure 8, bias-customized self-influence exhibits superior performance or is comparable to loss values, gradient norm, and self-influence. This observation is consistent with the result in the main paper.

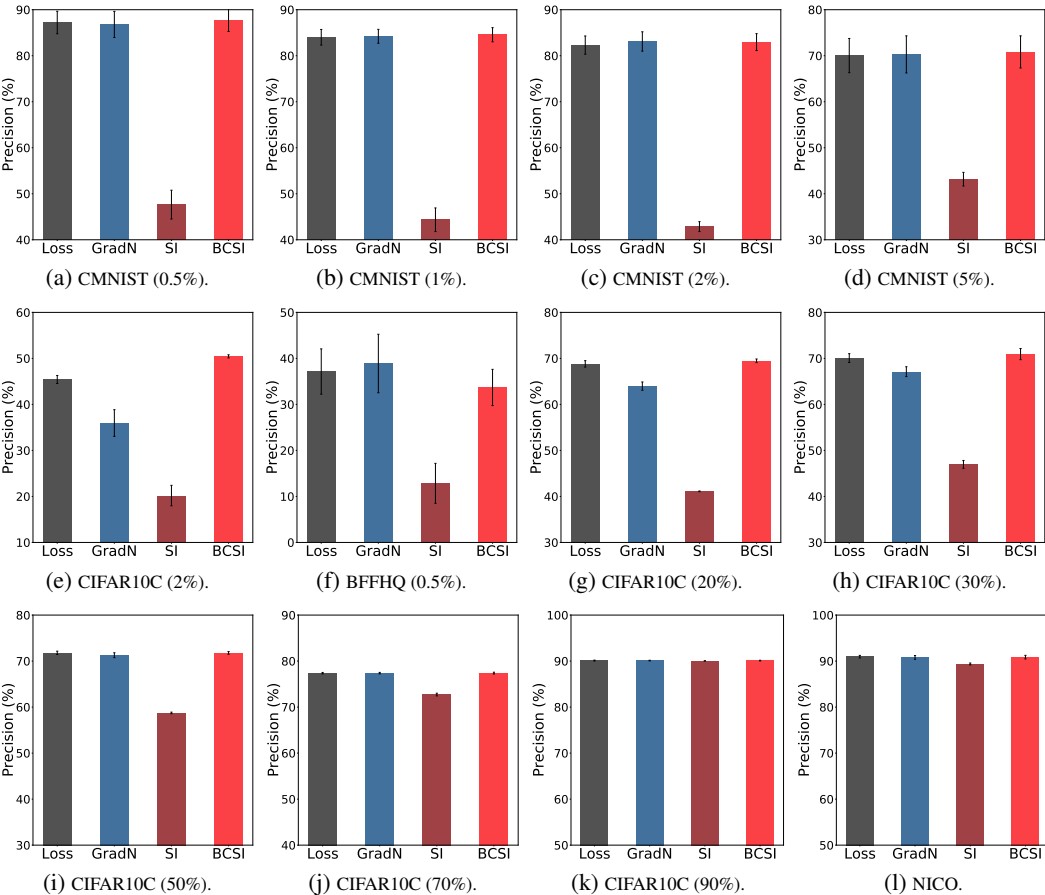

Figure 8: Comparison of bias-conflicting sample detection precisions between loss value, gradient norm (GradN), self-influence (SI), and bias-customized self-influence (BCSI) across various datasets. Gray, blue, dark red, and red bars indicate loss value, gradient norm, self-influence, and bias-customized self-influence, respectively.

## C  PERFORMANCE WITH RESPECT TO THE BIAS-CONFLICTING RATIO

In Figure 4.2 of the main paper, we showed the unbiased accuracy trends of the CIFAR10C dataset with respect to the bias-conflicting ratio for SelecMix and SelecMix with SePT. In Figure 9, we provide the CIFAR10C accuracy trends of LfF (Nam et al., 2020) and DFA (Lee et al., 2021) alone and with SePT.

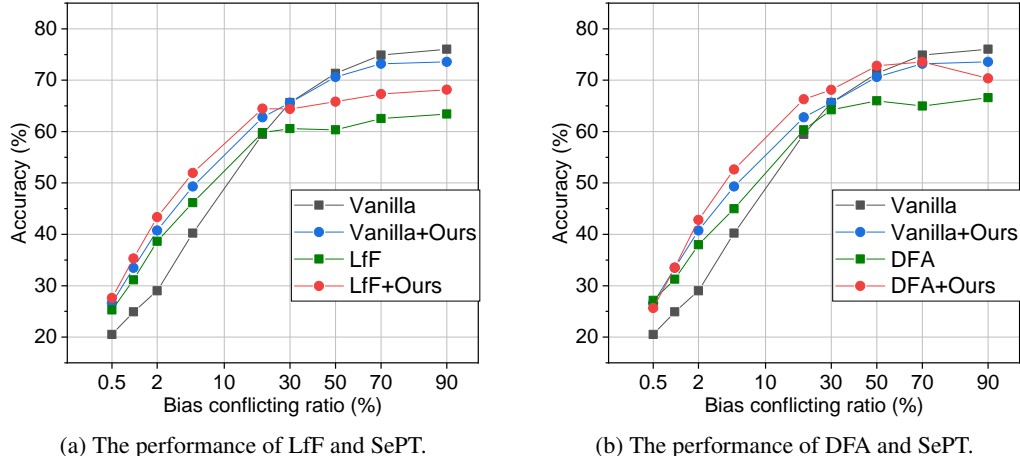

(a) The performance of LfF and SePT.       (b) The performance of DFA and SePT.

Figure 9: Performance of the other baselines and Ours (SePT) on the CIFAR10C dataset with varying bias ratio. The performance of LfF (Nam et al., 2020) is shown in Figure 9(a). Figure 9(b) displays the performance of DFA (Lee et al., 2021).

## D  BIAS-CONFLICTING RATIO OF THE PIVOTAL SET

We provide the resulting bias-conflicting ratios (*i.e.* bias-conflicting detection precisions) of the pivotal set produced by SePT across a variety of datasets. Table 4 and Table 5 show the bias-conflicting ratios for CMNIST (0.5%, 1%, 2%, 5%), CIFAR10C (0.5%, 1%, 2%, 5%, 20%, 30%, 50%, 70%), BFFHQ, Waterbirds, and NICO. In CMNIST, detection performance is inferior to other datasets. We conjecture that this result may be attributed to the significant estimation error of self-influence, which arises from using only the last layer in MLP. Note that we use MLP in CMNIST, following the prior works (Nam et al., 2020; Lee et al., 2021; Hwang et al., 2022).

Table 4: The average and the standard error of detection precision over three runs. Note that we compute the precision of the pivotal sets across varying ratios of bias-conflicting samples in CIFAR10C.

| | CIFAR10C | | | | | | | |
|---|---|---|---|---|---|---|---|---|
| | 0.5% | 1% | 2% | 5% | 20% | 30% | 50% | 70% |
| Accuracy | $45.57_{\pm 1.63}$ | $68.18_{\pm 0.96}$ | $86.13_{\pm 1.18}$ | $96.60_{\pm 0.11}$ | $99.94_{\pm 0.06}$ | $99.88_{\pm 0.12}$ | $98.30_{\pm 0.30}$ | $85.81_{\pm 2.05}$ |

Table 5: The average and the standard error of detection precision over three runs. Note that we compute the precision of the pivotal sets on CMNIST, BFFHQ, Waterbirds, and NICO.

| | CMNIST | | | | BFFHQ | Waterbirds | NICO |
|---|---|---|---|---|---|---|---|
| | 0.5% | 1% | 2% | 5% | 0.5% | 5% | |
| Accuracy | $56.68_{\pm 0.64}$ | $28.20_{\pm 0.58}$ | $4.33_{\pm 0.58}$ | $8.77_{\pm 9.47}$ | $37.60_{\pm 4.03}$ | $64.26_{\pm 1.93}$ | $94.01_{\pm 2.57}$ |

# E   COMPARISON OF TIME COSTS

In this section, we analyze the time cost of SePT and compare it with the other baselines. For a practical and tangible comparison, we measure the wall clock time for the CIFAR10C (0.5%) dataset. We run our experiments with a machine equipped with Intel Xeon Gold 5215 (Cascade Lake) processors, 252GB RAM, Nvidia GeForce RTX2080ti (11GB VRAM), and Samsung 860 PRO SSD. For self-influence calculation, we utilize the JAX (Bradbury et al., 2018) library for fast Hessian vector product calculation. For all other deep learning functionalities, we utilize Pytorch (Paszke et al., 2019). In Table 6, the wall-clock duration of each component of SePT is shown. We observe that the self-influence calculation step takes a longer time compared to the fine-tuning step due to the intersection process. However, this can be executed in parallel, which reduces the time cost of self-influence calculation approximately threefold. In Table 7, a wall-clock time comparison with the other baselines is shown. SePT consumes a significantly lesser amount of time, dropping to less than half the time of vanilla full training when the self-influence calculation is executed in parallel. Reflecting on these results, we assert that the time cost of SePT is rather small or even negligible compared to the full training time of other baselines.

Table 6: The average and the standard error of computational costs over three runs. We measure the computing time for full training as the wall-clock time of each component. Self-influence (parallel) represents calculating the bias-customized self-influence in GPU-parallel. Note that † indicates that corresponding methods use JAX while others utilize PyTorch.

| Component | Self-influence | Self-influence (parallel) | Fine-tuning |
|---|---|---|---|
| Time (min.) | $11.46^{\dagger}{}_{\pm0.08}$ | $3.86^{\dagger}{}_{\pm0.03}$ | $1.08_{\pm0.04}$ |

Table 7: The average and the standard error of computational costs over three runs. We measure the computing time for full training as the wall-clock time of each method. Ours (parallel) presents SePT which computes bias-customized self-influence in GPU-parallel. Note that † indicates that corresponding methods use JAX while others utilize PyTorch.

| Method | Vanilla | LfF | DFA | SelecMix | Ours | Ours (parallel) |
|---|---|---|---|---|---|---|
| Time (min.) | $22.55_{\pm0.32}$ | $33.64_{\pm0.34}$ | $53.18_{\pm2.55}$ | $352.53_{\pm5.13}$ | $12.54^{\dagger}{}_{\pm0.08}$ | $4.94^{\dagger}{}_{\pm0.03}$ |

# F   ANALYSIS FOR INTERSECTIONS OF PIVOTAL SETS

In this section, we analyze the effects of intersections between pivotal sets obtained from various random initializations of models. For the comparison, we provide the number of samples, detection precision, and performance after fine-tuning models across different numbers of the intersections in Table 8, Table 9, and Table 10. We observe that the detection precision increases as the number of intersections rises, while the number of samples in the pivotal set decreases. For the performance, a higher number of intersections shows effectiveness in the highly-biased scenarios, as bias-conflicting samples are scarce, and intersections reduce the size of the pivotal set. In contrast, a fewer intersections exhibit superior performance in low-based scenarios as there are abundant bias-conflicting samples. Note that, to observe the trend across varying ratios of bias-conflicting samples, we conduct experiments on CIFAR10C (0.5%, 1%, 2%, 5%, 20%, 30%, 50%, 70%).

Table 8: The average and the standard error of **the number of pivotal sets** over three runs considering numbers of intersections.

| Number of | CIFAR10C | | | | | | | |
|---|---|---|---|---|---|---|---|---|
| Intersections | 0.5% | 1% | 2% | 5% | 20% | 30% | 50% | 70% |
| 1 | 1000 | 1000 | 1000 | 1000 | 1000 | 1000 | 1000 | 1000 |
| 2 | $322.67_{\pm 3.38}$ | $386.67_{\pm 11.98}$ | $503.67_{\pm 39.75}$ | $577.00_{\pm 16.46}$ | $554.00_{\pm 63.38}$ | $421.67_{\pm 21.17}$ | $309.67_{\pm 40.03}$ | $290.00_{\pm 70.32}$ |
| 3 | $201.67_{\pm 4.91}$ | $267.00_{\pm 8.50}$ | $388.33_{\pm 19.06}$ | $430.33_{\pm 30.66}$ | $452.00_{\pm 65.09}$ | $281.00_{\pm 11.02}$ | $144.67_{\pm 30.99}$ | $141.67_{\pm 30.99}$ |

Table 9: The average and the standard error of **detection precision** over three runs considering numbers of intersections.

| Number of | CIFAR10C | | | | | | | |
|---|---|---|---|---|---|---|---|---|
| Intersections | 0.5% | 1% | 2% | 5% | 20% | 30% | 50% | 70% |
| 1 | $13.27_{\pm 0.50}$ | $24.90_{\pm 0.87}$ | $47.07_{\pm 0.65}$ | $76.27_{\pm 0.50}$ | $97.10_{\pm 0.95}$ | $97.60_{\pm 1.20}$ | $91.27_{\pm 2.02}$ | $80.83_{\pm 1.15}$ |
| 2 | $31.68_{\pm 1.61}$ | $52.83_{\pm 1.80}$ | $75.17_{\pm 3.66}$ | $92.01_{\pm 0.80}$ | $99.77_{\pm 0.16}$ | $99.02_{\pm 0.74}$ | $96.00_{\pm 0.47}$ | $83.68_{\pm 0.99}$ |
| 3 | $45.57_{\pm 1.63}$ | $68.18_{\pm 0.96}$ | $86.13_{\pm 1.18}$ | $96.60_{\pm 0.11}$ | $99.94_{\pm 0.06}$ | $99.88_{\pm 0.12}$ | $98.30_{\pm 0.30}$ | $85.81_{\pm 2.05}$ |

Table 10: The average and the standard error of **classification accuracy** of 'Ours+SelecMix' over three runs considering numbers of intersections.

| Number of | CIFAR10C | | | | | | | |
|---|---|---|---|---|---|---|---|---|
| Intersections | 0.5% | 1% | 2% | 5% | 20% | 30% | 50% | 70% |
| 1 | $36.44_{\pm 0.34}$ | $40.76_{\pm 0.03}$ | $49.57_{\pm 0.41}$ | $59.31_{\pm 0.15}$ | $\mathbf{67.99}_{\pm 0.33}$ | $\mathbf{67.04}_{\pm 0.65}$ | $\mathbf{67.39}_{\pm 0.79}$ | $\mathbf{70.09}_{\pm 0.28}$ |
| 2 | $\mathbf{38.85}_{\pm 0.62}$ | $43.47_{\pm 0.21}$ | $51.43_{\pm 0.53}$ | $\mathbf{60.22}_{\pm 0.19}$ | $66.96_{\pm 0.25}$ | $65.90_{\pm 0.81}$ | $66.77_{\pm 0.40}$ | $69.92_{\pm 0.53}$ |
| 3 | $38.74_{\pm 0.36}$ | $\mathbf{46.18}_{\pm 0.33}$ | $\mathbf{52.70}_{\pm 0.40}$ | $59.66_{\pm 0.31}$ | $66.66_{\pm 0.43}$ | $64.51_{\pm 1.44}$ | $66.45_{\pm 0.28}$ | $69.97_{\pm 0.21}$ |

## G ALGORITHM TABLES FOR SEPT

**Algorithm 1** Construct a pivotal set

1: **Input:** model parameters $\theta$, GCE $\mathcal{L}_{\text{GCE}}$, number of epochs $n_{\text{epoch}}$, learning rate $\rho$, number of classes $C$, train set $\mathbf{Z}$, number of topk $n_{\text{topk}}$
2: **Initialize:** Model parameter $\theta$.
3: **for** $i = 0, 1, 2, \cdots, n_{\text{epoch}}$ **do**
4:    $\theta \leftarrow \theta - \rho \nabla_\theta \mathcal{L}_{\text{GCE}}(\mathbf{Z}, \theta)$
5: **end for**
6: # Select samples with high self-influence
7: $\mathbf{Z}_{\text{P}} \leftarrow \emptyset$
8: **for** $c = 0, 1, 2, \cdots, C$ **do**
9:    $\mathbf{Z}_c \leftarrow \{(x, y) \in \mathbf{Z} | y = c\}$
10:    **for** $j = 0, 1, 2, \cdots, n_{\text{topk}}$ **do**
11:       $z_{\text{highest}} \leftarrow \arg\max_{z \in \mathbf{Z}_c} \mathcal{I}_{\text{self}}(z)$
12:       $\mathbf{Z}_c \leftarrow \mathbf{Z}_c \setminus \{z_{\text{highest}}\}$
13:       $\mathbf{Z}_{\text{P}} \leftarrow \mathbf{Z}_{\text{P}} \cup \{z_{\text{highest}}\}$
14:    **end for**
15: **end for**
16: **Output:** $\mathbf{Z}_{\text{P}}$

**Algorithm 2** Post-training with the pivotal set

1: **Input:** pre-trained model parameters $\theta^*$, CE $\mathcal{L}_{\text{CE}}$, number of iterations $n_{\text{iter}}$, learning rate $\rho$, train set $\mathbf{Z}$, pivotal set $\mathbf{Z}_{\text{P}}$, weight of remaining set $\lambda$
2: **Initialize:** Last-layer of model $\theta^*_{\text{last-layer}}$.
3: $\mathbf{Z}_{\text{R}} \leftarrow \mathbf{Z} \setminus \mathbf{Z}_{\text{P}}$
4: $n_{\text{P}} \leftarrow |\mathbf{Z}_{\text{P}}|$
5: **for** $i = 0, 1, 2, \cdots, n_{\text{iter}}$ **do**
6:    # Sample data from remaining samples
7:    $\mathbf{Z}_{\text{S}} \leftarrow \emptyset$
8:    **for** $j = 0, 1, 2, \cdots, n_{\text{P}}$ **do**
9:       $z \sim \mathbf{Z}_{\text{R}}$
10:       $\mathbf{Z}_{\text{S}} \leftarrow \mathbf{Z}_{\text{S}} \cup \{z\}$
11:    **end for**
12:    $\mathcal{L} \leftarrow \mathcal{L}_{\text{CE}}(\mathbf{Z}_{\text{P}}, \theta^*)$
13:    $\mathcal{L} \leftarrow \mathcal{L} + \lambda \mathcal{L}_{\text{CE}}(\mathbf{Z}_{\text{S}}, \theta^*)$
14:    $\theta^* \leftarrow \theta^* - \rho \nabla_\theta \mathcal{L}$
15: **end for**
16: **Output:** $\theta^*$

## H EXPERIMENTAL SETTINGS

### H.1 A DETAILED DESCRIPTION OF BENCHMARK DATASETS

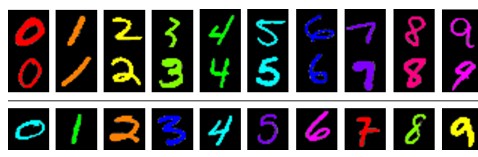
(a) Colored MNIST.

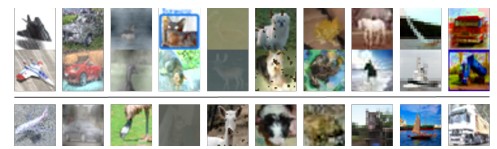
(b) Corrupted CIFAR10.

Figure 10: Example images of CMNIST and CIFAR10C. Images in the first and second rows are *bias-aligned* and images in the third row are *bias-conflicting*.
Figure 10(a) displays examples from the CMNIST dataset. Figure 10(b) displays examples from the CIFAR10C dataset.

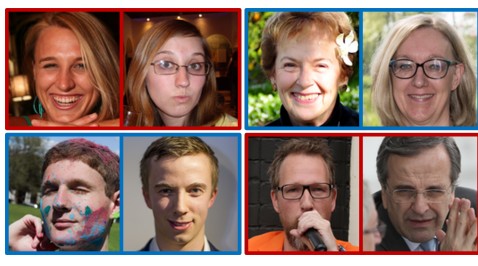
(a) Biased FFHQ.

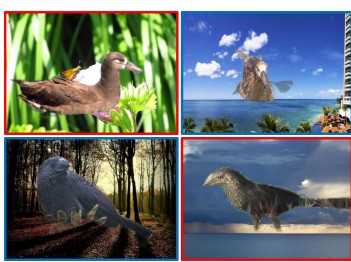
(b) Waterbirds.

Figure 11: Example images of BFFHQ and Waterbirds. The red-bordered images are *bias-aligned* and the blue-bordered images are *bias-conflicting*. Figure 11(a) displays examples from the BFFHQ dataset. Figure 11(b) displays examples from the Waterbirds dataset.

**Colored MNIST.** Colored MNIST (CMNIST) is a synthetically modified version of MNIST (Deng, 2012), where the digit is the label and the color is the bias. For example, an image of digit `0` is correlated with the color `red`. We use the following bias-conflicting ratios: $r \in \{0.5\%, 1\%, 2\%, 5\%\}$. The images are in `28 x 28` resolution and are resized to `32 x 32`.

There are approximately 55,000 training, 5,000 validation, and 10,000 test samples. Examples are shown in Figure 10(a).

**Corrupted CIFAR10.** Corrupted CIFAR10 (CIFAR10C) is a synthetically modified version of CIFAR10 (Krizhevsky et al., 2009) proposed by Hendrycks & Dietterich (2019) with the following common corruptions as the bias: {Snow, Frost, Fog, Brightness, Contrast, Spatter, Elastic transform, JPEG, Pixelate and Saturate}. We use the following bias-conflicting ratios: $r \in \{0.5\%, 1\%, 2\%, 5\%, 20\%, 30\%, 50\%, 90\%(\text{unbiased})\}$. The images are in 32 x 32 resolution. There are approximately 45,000 training, 5,000 validation, and 10,000 test samples. Examples are shown in Figure 10(b).

**Biased FFHQ.** Biased FFHQ (BFFHQ) (Lee et al., 2021) is a curated Flickr-Faces-HQ (FFHQ) (Karras et al., 2019) dataset, which consists of images of human faces. The designated task label is the age {young, old} while the bias attribute is the gender {man, woman}. The bias-conflicting ratio is $r \in \{0.5\%\}$. The images are in 128 x 128 resolution and are resized to 224 x 224. There are approximately 20,000 training, 1,000 validation, and 1,000 test samples. Examples are shown in Figure 11(a).

**Waterbirds.** Waterbirds is proposed by Sagawa et al. (2020), which synthetically combines bird images from the Caltech-UCSD Birds-200-2011 (CUB) with place background as bias. It consists of bird images to classify bird types {waterbird, landbird}, but their backgrounds {water, land} are correlated with bird types. The bias-conflicting ratio is $r \in \{5\%\}$. The images are in varying resolutions and are resized to 224 x 224. There are approximately 5,000 training, 1,000 validation, and 6,000 test samples. Examples are shown in Figure 11(b).

**NICO.** NICO is a dataset designed to evaluate non I.I.D. classification by simulating arbitrary distribution shifts. To evaluate debiasing methods, a subset composed of *animal* classes label is utilized, as in Wang et al. (2021). The class labels (*e.g.* "dog") are correlated to spurious contexts (*e.g.* "on grass", "in water", "in cage", "eating", "on beach", "lying", "running") which exhibits a long-tail distribution. The images are in varying resolutions and are resized to 224 x 224. There are approximately 3,000 training, 1,000 validation, and 1,000 test samples.

## H.2 BASELINES

We validate SePT by combining various debiasing approaches. Vanilla is the model trained by cross-entropy loss. GroupDRO (Sagawa et al., 2020) minimize the worst-group loss by exploiting group labels directly. ReBias (Bahng et al., 2020) trains models to be statistically independent of a given set of biased models, each of which encodes a distinct bias. LfF (Nam et al., 2020) detects bias-conflicting samples and allocates large loss weights on them. DFA (Lee et al., 2021) augments diverse features by swapping the features obtained from the biased model and concatenating the feature from the debiased model with the exchanged feature. BiaSwap (Kim et al., 2021) augments bias-conflicting samples by translating bias-aligned samples. BPA (Seo et al., 2022) utilizes a clustering method to identify pseudo-attributes using a clustering approach and adjusts loss weights according to the cluster size and its loss. SelecMix (Hwang et al., 2022) identifies and mixes a bias-contradicting pair within the same class while detecting and mixing a bias-aligned pair from different classes. Note that we adopt SelecMix+LfF rather than SelecMix since SelecMix+LfF exhibits superior performance than SelecMix (Hwang et al., 2022).

## H.3 EVALUATION PROTOCOL

We provide experimental setups for evaluation. In constructing pivotal sets, we adopt ResNet18 (He et al., 2016) as the base architecture for all datasets. For optimization, we employ the Adam optimizer (Kingma & Ba, 2015) with a learning rate of 0.001, and train the models for 5 epochs. To calculate self-influence, we only utilize the last layer of the models. In fine-tuning, we deploy ResNet18 for CMNIST, CIFAR10C, BFFHQ, and NICO while ResNet50 is used for Waterbirds as following other baselines (Nam et al., 2020; Lee et al., 2021; Liu et al., 2023). We adopt the Adam optimizer for CMNIST, CIFAR10C, BFFHQ, NICO while SGD is used for Waterbirds. For

the learning rate, we use 0.001 for CMNIST, CIFAR10C, Waterbirds, and $10^{-4}$ for BFFHQ. We apply cosine annealing (Loshchilov & Hutter, 2017) to decay the learning rate to $10^{-3}$ of the initial value. We utilize weight decay of $10^{-4}$ for all datasets . For baselines (Lee et al., 2021; Hwang et al., 2022), we use the officially released codes. For SePT, we adopt $k = 100, \lambda = 0.1$ for all datasets.

