# OpenReview forum: "Post-Training Recovery from Injected Bias with Self-Influence"
_ICLR.cc/2024/Conference — Submitted to ICLR 2024_

### Official Review · Reviewer_dWNy · 2023-10-23

**Soundness:** 2 fair
**Presentation:** 2 fair
**Contribution:** 3 good
**Rating:** 5
**Confidence:** 4

**Summary:**

This paper studies alleviation of spurious correlation caused by the bias in the data.
It proposes a fine-tuning framework to address such bias. Their approach leverages the idea of Influence Function (IF) to identify bias-conflicting samples. Specifically, by compute Self-Influence (SI), which measures a sample’s influence on the accuracy of itself, the framework can reveal data that hinder model generalization.

However, such a direct application of SI might not generate sufficient samples. Thus the paper further introduces bias-customized self-influence (BCSI), which effectively identifies bias-conflicting samples, with higher BCSI scores indicating pronounced conflicts with bias. They then create a pivotal subset that minimizes spurious correlations by considering BCSI scores of the training samples. Following this, fine-tuning of a biased model through a limited number of iterations is implemented using the selected subset of data, reducing injected bias. Experimental results show that the proposed method manifests effectiveness in low-bias scenarios.

**Strengths:**

1. This paper focuses on a central research problem in machine learning: spurious correlation/data bias.

2. This paper provides a fine-tuning method, which avoids the time-consuming retraining of a potential large model. This aligns with the current trend of research.

**Weaknesses:**

While I acknowledge the significance of the research question addressed in this paper and the presence of certain novel elements, I find the current presentation somewhat perplexing when attempting to decipher the results. Consequently, I have posed some specific questions below.
My current score is tentative, in the hope that the authors might provide additional clarifications.

**Questions:**

**1**.  Unclear definition in section 2.1: what is $s_y$ and $b_y$?
Does it indicate for a single class $y$, there can only be one relevant signal $s$, which is $s_y$, and only one irrelevant signal $b$, which is denoted as $b_y$? If yes, this seems to be the most extreme case of spurious correlation, i.e., the spurious correlation equals 1 for all class $y \in [C]$.
Or does it indicate that, for a single class, there can be one and only one spurious attribute? I.e., one class cannot be correlated with two spurious attributes?

**2**. In Figure 1, are the reported accuracies the validation accuracy measured on a validation set?

**3**. Figure 1(a), the pattern confused me. It seems that the precision is not monotonic w.r.t. The bias-conflicting sample ratio. Although the curve corresponding to 0.5% is at the top, the curve for 5% is above that of 1% and 2%. This makes me confused about the implication of this specific result and worried whether it is due to noise. It might be helpful to show the results averaged over multiple runs.

**4**. When the paper validated the proposed BCSI, it mentioned “In Figure 1(b), our approach exhibits consistent detection precision compared to conventional SI”. Could I ask why Figure 1(b) manifests “consistent detection”? It seems to be a large variation of range across bias ratios. I am confused.

**5**. From Figure 1e-1h, it seems that BCSI consistently outperformed others except for the extreme 0.5% case.

**6**. Regarding the last two paragraphs of Section 3.2: the paper claims that a drawback of many existing methods is that they cannot handle the setting with a lack of biased samples. To address this concern, the proposed solution is to add a simple cross entropy loss to the final loss function. My concern is, I did not see how this is novel compared to the existing methods. First, why cannot the previous methods also add a CE loss when the biased samples are lacking? Second, I am not convinced why adding a CE loss helps. When the pivotal set is small, isn’t it true that adding a randomly sample CE loss have close effect to ERM?

---

> ### Author Response · Authors · 2023-11-23
>
> We thank reviewer dWNy for the insightful comments. We address the reviewer's comments below.
>
> ---
>
> ### [Q1. Unclear definition in Section 2.1.]
>
> Our intention in describing our problem setting was to define the correlation between the task-relevant signal $s$ and the task-irrelevant bias $b$. We have revised our paper to take into account mild spurious correlations, as in $P(B=b_{y}|S=s_{y}) = p_{y}$, where $p_{y} \geq \frac{1}{C}$.
>
> ---
>
> ### [Q2. Reported accuracies in Figure 1.]
>
> The reported accuracies are the accuracies (precision) in detecting bias-conflicting samples within the training dataset.
>
> ---
>
> ### [Q3. Trend of precision w.r.t. the bias-conflicting sample ratio.]
>
> For bias ratios 1,2,5%, the detection precision of SI shows an increasing trend, as the absolute quantity of bias-conflicting samples increases (hence raising the lower bound of the precision). In the extreme bias case of 0.5%, the model fails to learn any task-related features due to the extreme scarcity of bias-conflicting samples. This phenomenon is observed in Figure 1(a) (now renamed to Figure 1(b)) (which is averaged over multiple runs - we removed their error bars due to graphical overlaps), where the detection precision does not decrease over epochs but plateaus, indicating that the model failed to learn task-relevant features even when trained for a sufficiently long time (200 epochs). For this specific case, the capability of SI to detect noisy label samples takes effect, resulting in the detection performance shown in Figure 1(e) (now renamed to Figure 2(a)).
>
> ---
>
> ### [Q4, Q5. Consistency of detection performance in Figure 1, and the extreme 0.5% case.]
>
> In extreme scenarios, like 0.5%, the task-related features of bias-conflicting samples significantly counteract the dominant features, notably malignant bias features, of the model thereby easily separable by SI. However, as shown in Figure 1(b) (now renamed to Figure 1(a)), there is a notable decline in the detection performance of SI as the ratio increases since the model has learned more of the task-related features of bias-conflicting samples for classification. To address the reviewer's concern, we revised the mentioned part to enhance clarity.
>
> ---
>
> ### [Q6. Novelty of the counterweight CE loss.]
>
> The focus of previous literature was to effectively learn under highly biased datasets by amplifying the presence of bias-conflicting samples. To this end, their main concern was to amplify the signal of bias-conflicting samples when training the model.
> However, we observe and raise the issue that such strategies underperform under relatively weak bias due to the under-utilization of the dataset as a whole for learning task-related features. Under this scenario, adding a CE loss aids the model in learning such features by utilizing the whole dataset. For SePT, we separately average the CE loss and the pivotal set loss before summation, so that it is agnostic to the relative size of the pivotal set and the balance between the two terms is maintained. We revised our paper to clarify this formulation.
>
> However, naively adding the CE loss to the formulation of previous methods such as [1, 2, 3] can cause detrimental effects to the learning process. Since they attempt to debias the model in the **training phase**, applying the CE loss destabilizes their learning process and injects bias, especially in the early stages of training where the model is most susceptible to bias[1]. In this sense, the naive introduction of the counterweight CE loss to previous works is not straightforward. However, SePT utilizes the CE loss during the **post-training recovery** phase, which is more stable compared to direct intervention in the training stage.
>
> ---
>
> [1] Nam et al. Learning from failure: De-biasing classifier from biased classifier. NeurIPS 2020.
>
> [2] Lee et al. Learning debiased representation via disentangled feature augmentation. NeurIPS 2021.
>
> [3] Hwang et al. Selecmix: Debiased learning by contradicting-pair sampling. NeurIPS 2022.

---

### Official Review · Reviewer_coZv · 2023-10-29

**Soundness:** 2 fair
**Presentation:** 3 good
**Contribution:** 2 fair
**Rating:** 3
**Confidence:** 2

**Summary:**

Biases in ML models may be caused by biased data samples. One type of samples, named $\textit{bias-conflicting samples}$ has been recognized and utilized as a pivotal subset with significantly diminished spurious correlations. However, the accuracy of detecting such $\textit{bias-conflicting samples}$ has been a challenging topic and inadequacy in this aspect may cause additional error in training. One of the major goals of this paper is to improve the preciseness in this task by introducing $\textit{self-influence score}$ to filter such $\textit{bias-conflicting samples}$ and therefore enables quick recovery of a biased model with minimal cost.

**Strengths:**

The paper addresses a critical issue, which is the preciseness of identifying bias-conflicting samples. The approach of using a quantitative measure (self-influence function) to identify such subset of samples is a novel approach, and this line of thoughts may be inspiring for designing more refined standard for this task.

**Weaknesses:**

The utility of this influence function is questionable. Although its definition is simple and standard in the area, I did not find its connections with biases and the issue of fairness overall. The goal of introducing this metric is to identify bias-conflicting data samples, and due to lack of connections with fairness, it is questionable whether this metric is well-aligned with the goal of this framework.

It would be great if the authors could provide more justifications on choosing this metric, either mathematically or conceptually.

**Questions:**

Please see the comments in the Weaknesses section.

---

> ### Author Response · Authors · 2023-11-23
>
> We thank the Reviewer coZv for constructive feedback. We address the reviewer's comments below.
>
> ---
>
> ### [W1. Connection of influence functions to bias and fairness.]
>
> First, we briefly explain the concept of using self-influence (SI) functions to identify mislabeled samples. A simple approach to estimate the influence of a single data point on a model's prediction is Leave-One-Out (LOO) retraining, which involves excluding a data point from the trainset, retraining the model, and assessing the impact by comparing the model's prediction before and after the exclusion. However, retraining for each data point is highly time-consuming work. To address the issue, Influence functions $\mathcal{I}(z_i,z_j)$ are proposed to approximate the LOO retraining to compute the influence of a data point. Typically, $z_j$ is sourced from the validation set when available. If not available, $z_j$ is set equal to $z_i$ which is called self-influence (SI). A higher SI score indicates more difficulty in prediction $z_j$ itself when excluding from the trainset. In other words, **the more $z_i$ counteracts the dominant features of the pre-trained model, the higher the SI scores it is measured**. In this context, SI has been previously used to identify mislabeled samples in the trainset.
>
> This property of SI establishes a connection for its usage in detecting bias-conflicting samples. Nonetheless, as shown in Figure 1, naively adopting SI for biased datasets generally fails due to the distinct nature of bias-conflicting samples compared to mislabeled samples. Bias-conflicting samples, containing **task-related features but under-prioritized** in training, differ from the dominant feature but do not counteract them. Consequently, models can learn task-relevant features in the late stage of training [2], leading to a reduction in the gap between the self-influences of bias-conflicting samples and those of bias-aligned samples. Therefore, we propose Bias-Conditioned Self-Influence (BCSI) to **restrict** the pre-trained model from learning task-related features of bias-conflicting samples by using Generalized Cross Entropy (GCE), and training the model for only five epochs based on the findings of Frankle et al.[1] the primary directions of the model's parameter weights had already been learned during iteration 500 to 2,000.
>
> Please note that to enhance readers' understanding, we provide a more comprehensive description in Section 3.1.
>
> ---
>
> [1] Frankle et al., "The Early Phase of Neural Network Training", ICLR, 2020.
>
> [2] Nam et al. "Learning from failure: De-biasing classifier from biased classifier." NeurIPS 2020.

---

### Official Review · Reviewer_pbA4 · 2023-10-31

**Soundness:** 1 poor
**Presentation:** 2 fair
**Contribution:** 2 fair
**Rating:** 3
**Confidence:** 3

**Summary:**

I think the paper works on a significant problem. I think their approach on influence functions on biased data could work. However, the weaknesses of the algorithms outweigh the strengths. First, the proposed approach is not motivated -- no idea why the modification is needed on self influence functions in biased dataset. Secondly, the novelty of the approach is not explained in the paper -- why the proposed approach will work intuitively, no theoretical support. There is some empirical support by showing some outperformance but I think it is not enough. So my decision is reject.

**Strengths:**

1. Training in a biased dataset is significant problem in machine learning domain. Influence functions have been used in removing noisy labels, detecting mislabeled samples etc. I think it is original to use influence functions in biased dataset.
2. The paper makes very good introduction of the paper, explains the biased dataset, influence functions in a very good way for a reader to follow to rest of the paper.

**Weaknesses:**

1. Figure 1 appears very early in the paper; it is very confusing, and not easy for any reader to follow. For example, what are the legends in Figure 1a. What is bias-aligned, bias-conflicting ? It is not explained anywhere before Figure 1 appears. It is also not very clear what Figure 1 tells us. Section 3.1 is supposed to explain why self influence does not work in biased data, but it is not clear first how it is applied in biased data, and then why it does not work and needs to be modified. As far as I can tell, Figure 1 only shows modified SI (which is not yet introduced in the paper till Section 3.1) outperforms SI but does not explain the fundamental reason of this outperformance.
2. The novelty of the proposed algorithm with respect to self-influence functions is not well-explained. Why training five epochs using GCE (what is GCE by the way ? ) and then using self influence function works but fully trained model does not ? There is also no theoretical backing of the proposed algorithm. If there is no theoretical backing of proposed approach, I would at least expect some intuitive explanation on why the proposed approach will work.

**Questions:**

1. How are you using self influence in biased data ?
2. Why does SI underperform in the biased data ?

---

> ### Author Response · Authors · 2023-11-23
>
> Thank you for your valuable feedback. We address the reviewer's concerns below.
>
> ---
>
> ### [Q1,2. W1,2.] How self-influence is used in detecting bias-conflicting samples, its limitations, and the motivations for our modifications.
>
> We agree with the need for a more detailed explanation to enhance readers' understanding of BCSI (Bias-Customized Self-Influence), and we provide a more comprehensive description in Section 3.1.
>
> The key point is that bias-conflicting samples possess correct task-related features, unlike mislabeled samples. Therefore, we have to restrict the model to learn the features of bias-conflicting samples to achieve better separation.
>
> Specifically, as we mentioned in the last paragraph of Section 2.2, self-influence is used for measuring the extent to which a sample **conflicts** with the **dominant feature** utilized by the pre-trained model for classification. However, as demonstrated in Figure 1, self-influence struggles to identify bias-conflicting samples in biased datasets. This mainly stems from the difference between the relationship of normal and mislabeled samples, and that of bias-aligned and bias-conflicting samples. While both mislabeled and bias-conflicting samples contrast the dominant features of the pre-trained model, they differ in nature.
>
> In the case of mislabeled samples, which are erroneously labeled as their name implies, they strongly **counteract** the dominant features of the pre-trained model thereby separable by self-influence. On the other hand, bias-conflicting samples, containing task-related features but merely **under-prioritized** in training, differ from the dominant features but do not counteract them. In other words, in a noisy labeled setting, the mislabeled sample's feature is incompatible with the dominant feature, whereas, in a biased setting, the bias-conflicting sample's feature is not only compatible but ideally, both should be utilized. This characteristic of bias-conflicting samples makes it harder for SI to separate bias-conflicting samples. As shown in Figure 1(b), as the ratio increases, detection performance declines since the model has learned more of the task-related features of bias-conflicting samples for classification.
>
> Motivated by this insight, we aim to restrict the model from learning task-related features to achieve better separation. To this end, we propose **BCSI** to effectively identify bias-conflicting samples. As mentioned in the third paragraph of Section 3.1, we used Generalized Cross Entropy (GCE) to train the model to become more biased, focusing less on the features of bias-conflicting samples. Furthermore, based on the findings of Frankle et al.[1] that the primary directions of the model's parameter weights had already been learned during the iteration 500 to 2,000, we train a model for only five epochs for masking the model focuses more on bias. Our method exhibits superior performance compared to conventional self-influence in general, as demonstrated in Figure 1 and Appendix A.
>
> ---
>
> ### [W1.1. Elaboration on Figure 1.]
>
> We have adjusted adjust the structural arrangement, including Figure 1, to improve readability. We will also provide a brief response to the questions.
>
> The terminology of bias-aligned and bias-conflicting is mentioned in Section 2.1 right above Figure 1. We will add a brief reference to the corresponding section in the caption of Figure 1.
> The legend in Figure 1(a) (now renamed to Figure 1(b)) denotes the proportion of bias-conflicting samples in the dataset. We added this information to the caption of Figure 1. Figure 1(a) illustrates that the detection accuracy of bias-conflicting samples via self-influence(SI) decreases as we use SI on biased models trained for longer epochs. Figure 1(b)-(h) (Figure 1(e)-(h) are now renamed to Figure 2(a)-(d)) demonstrates that our BCSI outperforms naive SI(save for the extreme case of 0.5%) as well as other metrics such as loss and gradient norm in identifying bias-conflicting samples.
>
> ---
>
> [1] Frankle et al., "The Early Phase of Neural Network Training", ICLR, 2020.

---

### Author Response · Authors · 2023-11-23

We thank all the reviewers for their insights. Based on their feedback, we revised our paper, where changes are colored in blue. The revised contents are as follows:

- We revised Section 2.2 and Section 3 to clarify our motivation for using self-influence to detect bias-conflicting samples.
- We revised Section 3 to clarify the motivation for BCSI, our proposed modification of SI to effectively detect bias-conflicting samples.
- For clarity of Figure 1, we divided Figure 1 into two figures(Figure 1(e)-(h) is renamed to Figure 2(a)-(d)), and additional details were added to their captions. Also, Figure 1(a) is renamed to Figure 1(b), and vice versa.
- We revised Section 2.1 to clarify an unclear definition.

Sincerely,

Authors

---

### Meta-Review · Area_Chair_KBLA · 2023-12-08

**Metareview:**

The authors develop an approach to reduce the bias introduced by spurious correlation into the training process of a neural network, avoiding the need for expensive retraining of the model several times. They develop a theoretical justification for their approach, and validate this empirically on a number of datasets. Reviewers brought up presentation issues with the paper that were adequately addressed in the rebuttal phase. However, there still remain unaddressed concerns regarding the lack of consistent improvements across experimental scenarios. I would encourage the authors to address the concerns raised and submit to a future venue.

**Justification For Why Not Higher Score:**

Lack of consistent significant improvements across experimental evaluations that are not well explained by the authors.

**Justification For Why Not Lower Score:**

N/A

---

### Decision · Program_Chairs · 2024-01-16

Reject